# Effect of montelukast in preventing dengue with warning signs among patients with dengue: A multicenter, randomized, double-blind, placebo-controlled trial

**Nattapat Nitinai**[1☯], **Worapong Nasomsong**[1☯], **Worayon Chuerboonchai**[2], **Akarawin Tweekittikul**[2], **Vutthikorn Khingmontri**[1,2], **Bawornnan Panuvatvanich**[1,2], **Thananut Bangchuad**[3], **Maynanchaya Pongpraijaroen**[1,3], **Tanapol Roongfa-ngarm**[1,3], **Vasin Vasikasin**[1,2] *

1 Department of Internal Medicine, Phramongkutklao Hospital and College of Medicine, Bangkok, Thailand, 2 Department of Internal Medicine, Ananda Mahidol Hospital, Lopburi, Thailand, 3 Department of Internal Medicine, Fort Suranari Hospital, Nakhon Ratchasima, Thailand

☯ These authors contributed equally to this work.
* vvasin@gmail.com

## Abstract

### Background

Montelukast has shown potential as a candidate treatment for dengue. This study aimed to evaluate the efficacy and safety of montelukast in preventing dengue with warning signs.

### Methods

This multicenter, randomized, double-blind, placebo-controlled trial enrolled adult participants with NS1 antigenemia in Thailand. The participants were randomly assigned to receive either oral montelukast (10 mg) or a placebo for 10 days or until all symptoms resolved.

### Results

Between January 2021 and June 2023, 358 participants were enrolled and randomly assigned (1:1) to receive either montelukast or placebo. The incidence rate of warning signs in the montelukast group and the placebo group were 9.5% and 7.8% per day, respectively. There was no difference between the two groups (HR 1.36; 95%CI 0.94–1.96, P = 0.105). No statistically significant differences were observed in the incidence rate of severe dengue, hemoconcentration, thrombocytopenia, admission, or recovery from dengue. Neither dengue shock, nor mortality occurred. The montelukast group exhibited a decreased incidence rate of transaminase elevations (0.7% vs 1.4% per day, HR: 0.48, 95%CI 0.25–0.90, P = 0.023).

### Conclusion

Oral montelukast does not reduce the incidence of warning signs among patients with dengue. Nevertheless, the observed decrease in transaminase elevations warrants further investigation to evaluate the potential effect of montelukast.

**Data Availability Statement:** The data cannot be shared publicly because the informed consent did not include the permission. De-identified data can be requested by researchers who meet the criteria for access confidential data for use in independent scientific research. Requests should be sent to Department of Medicine, Phramongkutklao Hospital Data Access Committee (contact via pmkmedicine@live.com or Jantima Traipattanakul [jantimajob@pcm.ac.th]).

**Funding:** This study was supported by the Royal College of Physicians of Thailand (8/2564 to VV) and Phramongkutklao College of Medicine. The funders had no role in study design, data collection and analysis, decision to publish, or preparation of the manuscript.

**Competing interests:** The authors have declared that no competing interests exist.

## Clinical Trials Registration

Clinicaltrials.gov, NCT04673422, registered on 9 December 2020

## Author summary

Although there is no specific treatment for dengue to date, montelukast shows promise as a potential therapeutic agent. An open-label study previously showed a 71% reduction in dengue shock among patients with dengue who were administered montelukast. However, concerns were raised regarding the clinical study's lack of a power analysis or predefined statistical difference. Because of its affordability, relatively safe profile, and widespread availability, exploring its role in preventing dengue complications holds great significance. This study is a double-blind clinical trial with pragmatic design assessing the efficacy of montelukast in dengue. In contrast to the previous open-label study, montelukast does not reduce the incidence of warning signs among patients with dengue. Nevertheless, a reduction in the incidence of transaminase elevations was observed. In conclusion, montelukast should not be used for the prevention of warning signs in patients with dengue. However, the observed decrease in transaminase elevations warrants further investigation to evaluate the potential effect of montelukast.

## Introduction

Dengue poses a growing public health challenge in tropical countries [1]. With an estimated four billion people at risk, it leads to significant mortality [2].The primary cause of mortality in dengue is vascular leak syndrome, which manifests with varying severity, from mild cases to the life-threatening dengue shock syndrome [3].

Mast cells have recently been proposed as a potentially important regulator for vascular leakage in dengue [4–7]. When activated, mast cells release various cytokines, including vascular endothelial growth factor, chymase, tryptase, histamine, and leukotriene, all playing roles in increasing vascular permeability. Research suggests that mast cell stabilizers or antihistamine drugs might help manage this condition [4].

Leukotriene might play an important role in promoting plasma leakage and leukocyte adhesion in postcapillary venules [8]. In dengue patients, leukotriene levels elevate during the febrile and defervescence stage of the baseline values, and return to baseline in the convalescence stage [9]. Elevated leukotriene levels during the first four days of illness are associated with a higher risk of developing dengue hemorrhagic fever [10].

Montelukast is a leukotriene receptor antagonist for the treatment of allergic disorders. It possesses the capability to inhibit various leukotriene receptors to stabilize mast cells [11]. Animal model studies found that montelukast could reduce plasma leakage [5]. Dengue infected mice administered montelukast displayed a notable decrease in Evans blue dye leakage into their tissues compared to untreated mice. Notably, Montelukast exhibits potential effects beyond addressing vascular leakage; it has been observed to impede the dengue virus infection of lymphoid cells mediated by exogenous LTE4 [12].

A 2018 open-label retrospectively registered study found that patients treated with montelukast 10 mg once daily had a 22% absolute risk reduction in dengue shock syndrome compared to standard treatment.[13] However, to date, no double-blind pre-registered randomized controlled trial has evaluated the efficacy of montelukast in dengue patients.

The current management of dengue primarily involves symptomatic treatment and intravenous fluid replacement, with no specific treatment demonstrating clear benefits in preventing complications [3]. Given that montelukast is an affordable, relatively safe, and widely available medication, investigating its potential in preventing dengue complications is of paramount importance.

This study aimed to provide data on the efficacy and safety of a 10-day course of oral 10-mg montelukast in preventing warning signs in dengue infection.

## Methods

### Ethical statement

This study was conducted in accordance with Good Clinical Practice Guidelines and followed the approved protocol, including any subsequent amendments, as reviewed and sanctioned by the Institutional Review Board, of both Royal Thai Army Medical Department (www.irbrta-research.com reference number S077h/63) and the involved hospitals. The safety of the trial was overseen by the Research Unit at Phramongkutklao Hospital. All participants provided written informed consent before being enrolled.

### Study design

This study was a multicenter, parallel-group, randomized, double-blind, placebo-controlled clinical trial. The study enrolled patients from three Royal Thai Army hospitals situated in Thailand, namely Phramongkutklao, Fort Suranari, and Ananda Mahidol Hospitals. These hospitals had a capacity of 1,200, 420, and 321 beds, respectively.

The data and safety monitoring board performed a formal interim analysis after the enrolment of 50 and 75% of participants. Data collection and management were performed by the investigators and associated research personnel, while the final analyses were carried out by the trial statistician.

### Participants

Eligible patients were individuals aged 20 years or older, who were diagnosed with a clinical diagnosis of dengue infection, as confirmed by either positive NS1 antigen (Standard F Dengue NS1 Ag FIA, SD Biosensor, Gyeonggi-do, Republic of Korea) or positive PCR test (VIASURE, CerTest Biotec, Zaragoza, Spain), irrespective of the duration of illness. Exclusion criteria included the presence of any warning signs of dengue, concurrent diagnosis of other fever-causing conditions, such as malaria or heat stroke, pregnancy, inability to take medication orally, critical illness necessitating intubation or admission to an intensive care unit, inability to communicate, or any indication of montelukast. The warning signs for exclusion derived from the WHO 2009 guidelines [14], and included abdominal tenderness or pain, persistent vomiting, clinical fluid accumulation, mucosal bleeding, liver enlargement exceeding 2 cm. However, due to the enrolment timing of most participants during their initial hospital visit, the increase in hematocrit concurrent with a decrease in platelet count could not be determined at that stage. Consequently, the exclusion criteria based on laboratory results relied on the WHO 1997 guidelines definition of dengue hemorrhagic fever [15]. Eligible patients with a hematocrit exceeding 50% of the whole blood volume and platelet count below $100\times10^9$/L were excluded from the study. All participants were followed up for a period of up to 14 days.

### Randomization and masking

Upon enrolment, participants were randomly assigned, in a 1:1 ratio, to receive either montelukast or placebo. A computer-generated randomization list was prepared by an independent

statistician. The list was stratified by the center and used a block size of four. Each center received the study medications, the indistinguishable 10-mg tablets packaged in sequentially numbered containers. These medications were supplied by Unison Laboratories (Chachoengsao, Thailand).

Once a participant met all the necessary entry criteria, they were assigned the next available randomization number in chronological order at their respective center. The participants, investigators, and medical teams involved in the study were kept unaware of the group assignments. Interim analyses were performed masked to treatment allocation.

## Procedures

Researchers were notified regarding the eligible patients by the laboratory professionals who routinely conducted NS1 antigen or PCR tests on patient specimens. Within 24 hours of the positive results, these eligible patients were approached for participation. They were then administered either 10 mg of montelukast or a placebo orally immediately and continued to take it daily for 10 days or until they recovered, whichever came first. Recovery was defined as the point when attending physicians confirmed the discontinuation of follow-up appointments.

If any participant developed a convalescence rash indicating disease recovery, they were advised to stop taking the medication. Convalescence rash was defined as any rash that appeared after the fever had subsided. Only new rashes that occurred during the febrile phase were considered potential adverse drug reactions.

The participants were followed at least every other day for 14 days or until their attending physicians ended the follow-up appointments, whichever is shorter. The follow-up parameters included clinical symptoms, warning signs, potential adverse drug reactions, hospitalization status, death status, and laboratory results, irrespective of whether the participants completed the full scheduled course of the allocated study treatment. Mandatory laboratory investigation at each visit was complete blood count. Study staff sought follow-up information through telephone calls and reviewing information from medical notes and healthcare systems.

## Outcomes

The primary outcome was the incidence rate of participants with the first time of any warning signs, as per WHO guidelines [14]. These warning signs included abdominal tenderness or pain, persistent vomiting, clinical fluid accumulation, mucosal bleeding, liver enlargement greater than 2cm, and increasing hematocrit with decreasing platelet levels from the time of enrolment. However, subjective lethargy was excluded as a criterion for warning signs it had low specificity to severe dengue [16].

Secondary outcomes assessed in the intention-to-treat population included the incidence rate of severe dengue, decreased platelet count, increased hematocrit, hospitalization, transaminase elevations, and disease recovery.

The severe dengue was defined according to WHO guidelines [14]. Transaminase elevations were defined as a combination of aspartate transaminase (AST) and alanine transaminase (ALT) concentrations of more than 3 times the upper limit of normal, excluding those who had transaminase elevations at enrolment [17].

Safety was evaluated by use of CTCAE (version 5), excluding rash compatible with convalescence rash.

## Statistical analysis

Based on findings in a study reporting the incidence of dengue with warning signs excluding lethargy [16], we assumed a 54.5% event rate for the primary outcome in the placebo group.

Regarding the 22% absolute risk reduction in dengue shock syndrome in an open-label study [13], we assumed a 15% effect size for the intervention (absolute risk reduction of 0.15). With a 5% rate of loss to follow-up, and a two-sided test at the 5% level, a sample size of 358 provides approximately 80% power to detect a reduction in the primary endpoint between the two arms.

Data collection was conducted using case record forms, which were later transferred to an Excel spreadsheet via email. Validated data were transferred to R program for statistical analyses.

The analysis approach involved intention-to-treat analyses, where all participants randomized to different treatment arms were compared, regardless of whether they received their allocated treatment. For comparisons between groups, Pearson's chi-square test or Fisher's exact test was used for proportions, and Student's T-test or Mann-Whitney U test was used for continuous outcomes, as appropriate.

To estimate cumulative incidence and assess the efficacy of montelukast compared to placebo, the Kaplan–Meier method and the stratified log-rank statistic were employed. Additionally, an adjusted analysis of survival was conducted using a Cox proportional-hazards regression model that included baseline variables with a p value of 0.10 or less for between-group comparisons.

The generalized estimating equation (GEE) for global tests of repeated measures was used to examine the association of various laboratory results and body temperature with treatment groups and days until peak. The days until peak were adjusted for the days of the minimum platelet, white blood cell count, body temperature, or maximum hematocrit, AST, ALT of each participant with various stages of illness.

Interim analyses for safety and efficacy at 50 and 75% of information were performed by the independent statistician from an independent data and safety monitoring board with group sequential stopping boundaries defined with the use of a Lan-DeMets spending function with O'Brien–Fleming monitoring boundary. The trial would not be stopped in case of futility [18].

All statistical analyses were performed with R program, version 4.3.1 for statistical analyses. All p values were two-tailed. This trial was registered at ClinicalTrials.gov with the study identifier: NCT04673422.

## Results

### Population of patients

Between January 15, 2021 and June 17, 2023, a total of 406 eligible patients with NS1 antigenemia were identified at the study sites. Out of these, 358 participants were enrolled and randomly assigned to receive either montelukast (n = 179) or placebo (n = 179) within 24 hours after detecting dengue NS1 antigen (Fig 1). Among the 358 enrolled participants, one person from the placebo group was excluded owing to an inclusion mistake (baseline hematocrit of 50.3%). This participant did not receive any study medication and was withdrawn from the trial within 24 hours of enrolment. As a result, 357 participants were included in the intention-to-treat analysis.

A total of 25 participants were lost to follow up and one participant did not take any medication, leaving 163 participants in montelukast group and 168 participants in placebo group included in per-protocol analysis.

Descriptive baseline characteristics are presented in Table 1. The median age was 23 (IQR 22–29) years, with 303 (84.6%) of them being male. The median duration of fever before enrolment was 2.46 (IQR 1.50–3.67) days. The most prevalent serotypes were serotype 1 (236,

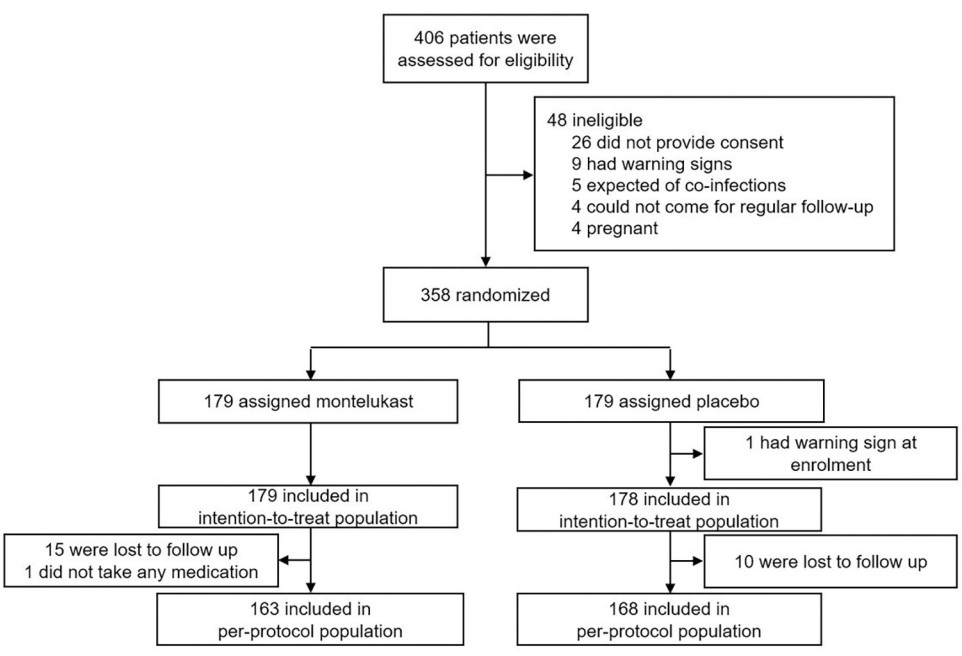

**Fig 1. Trial profile.**

67.8%), and serotype 2 (81, 23.3%). The participants took the study medications for a median of 5 (IQR 4–7) days.

## Clinical outcomes

The cumulative incidences of warning signs in the montelukast group and the placebo group were 35.2% (63/179) and 29.8% (53/178), respectively. Almost all participants (113/116, 97.4%) experienced increasing hematocrit with decreasing platelet counts as their first warning signs. In two participants, severe dengue preceded laboratory abnormalities. One participant experienced mucosal bleeding without an increase in hematocrit and decrease in platelet count.

The incidence rate of warning signs in the montelukast group and the placebo group was 9.5% and 7.8% per day, respectively. There was no difference between the two groups (HR 1.36, 95%CI 0.94–1.96, P = 0.105), as shown in Fig 2 and Table 2.

A total of 58/275 (21.1%) participants were admitted after enrolment. The median length of hospital stay was 3 (IQR 2–4.5) days. Severe dengue was found in 5/377 (2.5%) participants, all of whom had AST levels exceeding 1,000 U/L. One participant in the placebo group also experienced upper gastrointestinal tract hemorrhage.

There were no statistically significant differences in the incidence rate of severe dengue, increased hematocrit, decreased platelet, admission, or recovery from dengue, as shown in Table 2. There was no difference in the length of hospital stay (median 3 (IQR: 2–4) vs 3 (IQR: 2–5) days, p value = 0.478). No dengue shock or mortality observed in this study.

Transaminase levels were evaluated in 68.3% of participants (244/357); 65.4% (117/179) and 71.3% (127/178) in the montelukast and placebo group, respectively. There was a difference in the incidence rate of transaminase elevations. The cumulative incidence of transaminase elevation in the montelukast group and the placebo group was at 8.1% (14/172) and 16.7% (29/174), respectively. The incidence rate in the montelukast group and placebo group was 0.7% and 1.4% per day, respectively (HR: 0.48, 95%CI 0.25–0.90, P = 0.023), as depicted in

**Table 1. Baseline characteristics.**

| Characteristic | Montelukast n = 179 | Placebo n = 178 |
|---|---|---|
| Male (%) | 154 (86.0) | 148 (83.1) |
| Age (years) | 23 (22–28) | 23 (22–30) |
| Fever duration (days) | 2.4 (1.4–3.7) | 2.6 (1.5–3.7) |
| Body temperature (˚C) | 38.0 (37.0–38.7) | 37.8 (37.0–38.5) |
| Body weight (kg) | 67 (60–76) | 68 (60–80) |
| Underlying diseases | 10 (5.6) | 20 (11.2) |
| • Hypertension | 2 (1.1) | 10 (5.6) |
| • Diabetes | 2 (1.1) | 5 (2.8) |
| • Hyperlipidemia | 2 (1.1) | 4 (2.2) |
| • Coronary and cerebrovascular diseases | 1 (0.6) | 2 (1.1) |
| • Others† | 3 (1.7) | 4 (2.2) |
| Site | | |
| • Phramongkutklao Hospital | 169 (94.4) | 168 (94.9) |
| • Ananda Mahidol Hospital | 8 (4.5) | 8 (4.5) |
| • Fort Suranari Hospital | 2 (1.1) | 1 (0.6) |
| Hematocrit (%) | 43.2 (40.7–45.3) | 42.1 (40.3–45.3) |
| Leukocyte ($\times 10^6$/L) | 3.9 (2.9–5.2) | 3.9 (3.1–5.4) |
| Platelet ($\times 10^9$/L) | 144 (110–198) | 153 (119–184.5) |
| AST (U/L) | 56.3 (34.9–94.4) | 50.9 (34.9–94.3) |
| ALT (U/L) | 34.9 (21.7–75.8) | 35.6 (21–66) |
| Creatinine (mg/dL) | 1.1 (1–1.2) | 1.0 (0.8–1.2) |
| Admission | 26 (14.5) | 38 (21.3) |
| Serotype | | |
| • Serotype 1 | 109 (62.6) | 127 (73.0) |
| • Serotype 2 | 47 (27.0) | 34 (19.5) |
| • Serotype 3 | 2 (1.1) | 3 (1.7) |
| • Serotype 4 | 17 (9.8) | 11 (6.3) |

Data are n (%), or median (IQR).

† Other underlying diseases comprised: montelukast– 1 chronic kidney disease, 1 migraine, 1 osteoarthritis of knee; placebo– 1 deep venous thrombosis, 1 asthma, 1 vitiligo, 1 hypothyroidism.

Fig 3. S1–S4 Figs illustrated the measured transaminase levels of all individuals. GEE analysis showed significantly lower estimate of AST (-87.88 U/L, p = 0.023) and trend toward lower estimate of ALT (-31.04 U/L, p = 0.078) in the montelukast group. For further laboratory results and body temperature, refer to S5–S8 Figs and S1 Table.

## Safety

No serious adverse event was found in both groups. Potential adverse drug reactions were found in three participants, all of whom were in placebo group (P = 0.244). These reactions included two cases of dizziness (grade 2), and one case of oedema face (grade 1). There were no reports of any rash development before defervescence in either group.

## Subgroup analyses

No differences in the primary outcome were found between the two groups in per-protocol analysis, or any subgroups regardless of study sites, serotypes, sex, underlying diseases, or admission at enrolment. The findings of these analyses are presented in Fig 4.

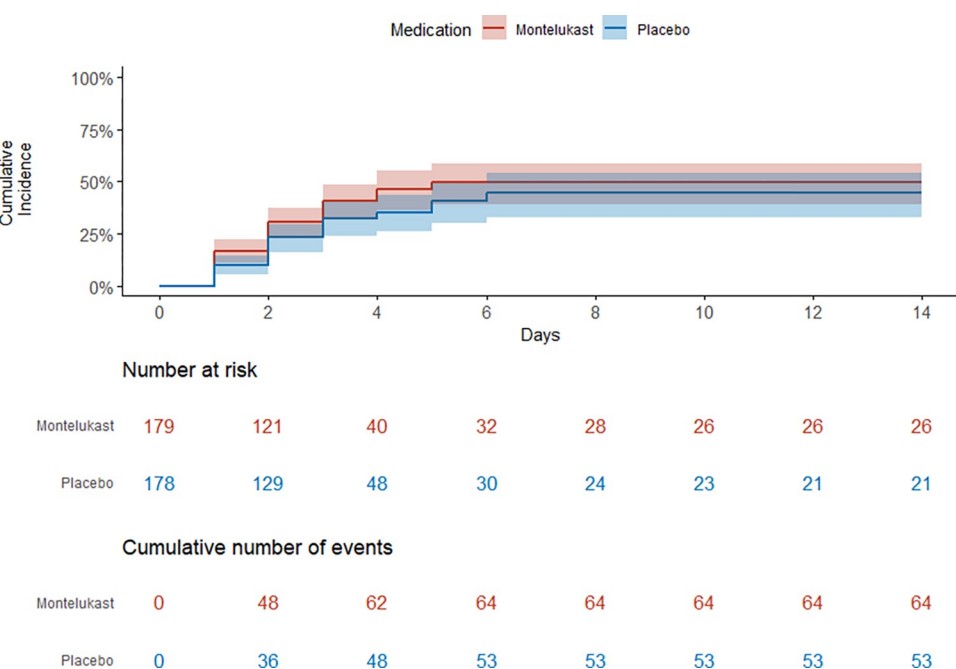

**Fig 2. Primary outcome.** Kaplan-Meier curve shows the number of participants at risk and the cumulative number of events in the whole study cohort. Shaded regions around curves represent 95% CIs.

## Discussion

In our study, the findings revealed that montelukast did not have efficacy in reducing the incidence of dengue with warning signs, severe dengue, hemoconcentration, thrombocytopenia, hospitalization, or disease recovery. On the other hand, there was a lower incidence rate of transaminase elevations in the montelukast group.

The cumulative incidence of warning signs was lower than what was expected with 32.5% observed versus 54.5% expected. This could possibly be attributed to the fact that a majority of the participants were young and previously healthy. Additionally, the exclusion of patients with warning signs at baseline which led to 2.2% of eligible patients being excluded may have contributed to this lower incidence. Nonetheless, this lower incidence should not have

**Table 2. Outcomes.**

| Outcome | Montelukast | | Placebo | | Hazard Ratio (95% CI) | p value |
|---|---|---|---|---|---|---|
| | n | % per day | n | % per day | | |
| **All participants** | n = 179 | | n = 178 | | | |
| **Primary outcome** | | | | | | |
| • Warning signs | 64 | 9.5% | 53 | 7.8% | 1.36 (0.94–1.96) | 0.105 |
| **Secondary outcomes** | | | | | | |
| • Severe dengue | 2 | 0.1% | 3 | 0.1% | 0.64 (0.11–3.81) | 0.621 |
| • Decreased platelet count | 81 | 4.9% | 76 | 4.4% | 1.26 (0.91–1.73) | 0.162 |
| • Increased hematocrit | 122 | 8.7% | 123 | 9.3% | 1.10 (0.85–1.41) | 0.478 |
| • Admission | 25 | 1.5% | 33 | 2.3% | 0.73 (0.44–1.24) | 0.247 |
| • Transaminase elevations | 14 | 0.7% | 29 | 1.4% | 0.48 (0.25–0.90) | 0.023 |
| • Recovery | 179 | 8.8% | 178 | 9.6% | 1.17 (0.95–1.44) | 0.149 |

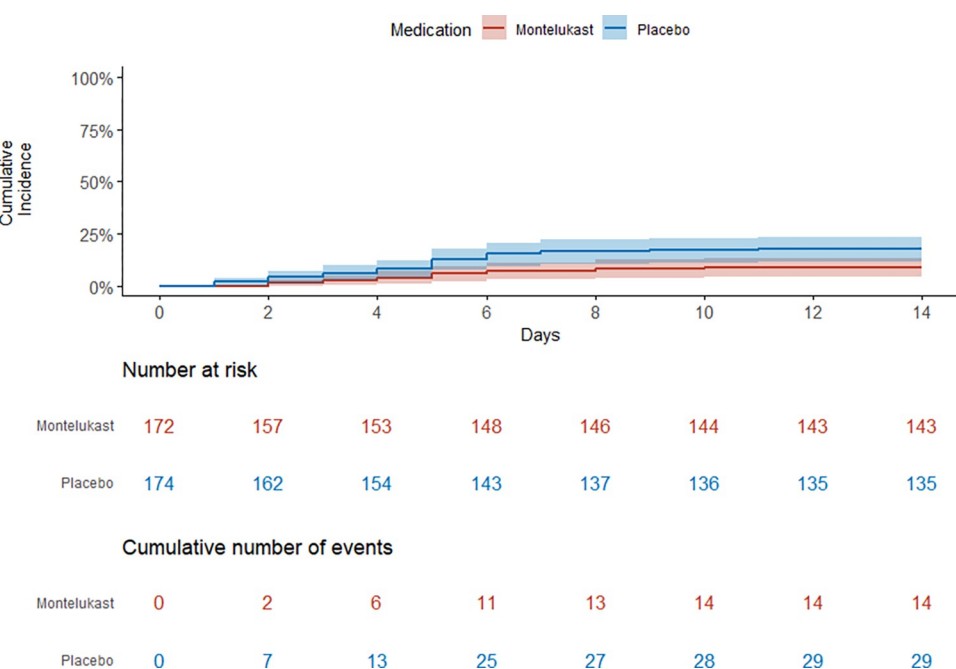

**Fig 3. Incidence of transaminase elevation overtime.** Kaplan-Meier curve shows the number of participants at risk and the cumulative number of events in the whole study cohort. Shaded regions around curves represent 95% CIs.

underestimated montelukast efficacy since there was no observed trend toward improvement in efficacy.

The findings contrast with the previous open-label study that showed the beneficial outcome with montelukast in reducing the incidence of dengue shock syndrome [13]. Several factors may account for these divergent results. Firstly, different modalities of supportive

| Subgroup | No. of patients | | Hazard Ratio | 95%CI | |
|---|---|---|---|---|---|
| Overall | 357 | | 1.36 | 0.94 | 1.96 |
| Per-protocol population | 331 | | 1.32 | 0.91 | 1.91 |
| *Study site* | | | | | |
| Phramongkutklao Hospital | 338 | | 1.35 | 0.93 | 1.95 |
| Other hospitals | 19 | | 2.46 | 0.09 | 64.62 |
| *Serotype* | | | | | |
| Serotype 1 | 236 | | 1.41 | 0.88 | 2.27 |
| Other serotypes | 111 | | 1.12 | 0.62 | 2.04 |
| *Sex* | | | | | |
| Male | 302 | | 1.38 | 0.92 | 2.06 |
| Female | 55 | | 1.43 | 0.56 | 3.64 |
| *Underlying diseases* | | | | | |
| Yes | 30 | | 1.32 | 0.90 | 1.95 |
| No | 327 | | 1.52 | 0.48 | 4.80 |
| *Admission at enrolment* | | | | | |
| Yes | 64 | | 1.16 | 0.53 | 2.54 |
| No | 293 | | 1.44 | 0.95 | 2.19 |

0.1 1 10 100

← Montelukast better Placebo better →

**Fig 4. Subgroup analyses for the primary outcome.** CI: Confidence interval.

treatments in the two studies may have influenced the outcomes. Secondly, the absence of dengue shock syndrome in our study, which vastly differed from the referred study, may be due to the majority of participants being young and healthy. Warning signs closely monitored and managed in this trial might also prevent the development of dengue shock syndrome. Finally, our study had less ascertainment bias due to the use of a pre-registered double-blind method.

Nevertheless, our study revealed a lower incidence rate of transaminase elevations in the montelukast group compared to the placebo group. Although not all participants had their transaminase levels measured, the cumulative incidence in the placebo group (16.7%) was comparable to two previous studies where transaminase levels were monitored daily (16.3% and 17.9%) [17,19]. While this finding could be a chance occurrence, it is crucial to explore the possible mechanism, given that severe acute hepatitis in dengue is associated with poor outcomes such as prolonged hospital stay, bleeding, renal failure, and mortality [20,21].

The mechanism behind transaminase elevations is attributed to hepatic cell apoptosis caused by viral cytopathy, hypoxic mitochondrial dysfunction, immune response, and accelerated endoplasmic reticular stress [22]. As part of the host immune response, various cytokines, including interleukin (IL)-2, IL-6, tumor necrosis factor (TNF)-α, and interferon (IFN)-γ, contribute to severe dengue [22]. Other mechanisms for severe dengue involve platelet-activating factor and serum phospholipase A2 secreted by mast cells [23]. All of these factors may contribute to hepatic cell apoptosis. However, no direct correlations have been found between leukotriene levels and hepatic cell apoptosis.

It is worth noting that the intake of paracetamol in dengue has been confirmed to increase the incidence of transaminase elevations [17]. However, the difference in unrecorded paracetamol intake between the two groups is unlikely to be the reason, as the duration of illness was similar in both groups. The trend toward higher body temperature was observed in the montelukast group (p = 0.065). Further research in in-patient setting, where transaminase levels and other clinical parameters can be closely monitored, may be needed to confirm the observed differences in transaminase elevations between the montelukast and placebo groups.

Our study had certain limitations. Firstly, the medication adherence of individual participants could not be documented reliably, despite attempting to notify and check drug compliance through telephone calls. Secondly, the majority of participants were young Thai men, which may not fully represent real-world practice. Thirdly, because of the brief two-year trial duration, there was limited diversity in the prevalence of serotypes, with serotype 1 comprising two-thirds of the participants. The efficacy might not extend to other serotypes. However, subgroup analysis did not reveal any sign of montelukast benefit. Finally, attending physicians had considerable freedom in conducting laboratory investigations due to the limited resources. Although this approach was pragmatic, it might introduce the possibility of unaccountable bias influencing decision-making for investigation. Nevertheless, objective outcomes such as admission rates and severe dengue were comparable between the two groups.

In conclusion, the 10-day course of 10-mg montelukast does not prevent dengue with warning signs compared to a placebo. Interestingly, the montelukast group showed a lower incidence rate of transaminase elevations compared to the placebo group. Further research is required to thoroughly evaluate this potential effect.

## Supporting information

**S1 CONSORT Checklist. CONSORT Checklist.**
(PDF)

**S1 Protocol. Study Protocol.**
(PDF)

**S1 Fig. AST concentrations of all individuals and the mean concentration of each group with 95%CI on each day after enrolment.**
(JPG)

**S2 Fig. ALT concentrations of all individuals and the mean concentration of each group with 95%CI on each day after enrolment.**
(JPG)

**S3 Fig. AST concentrations of all individuals and the mean concentration of each group with 95%CI on each day before maximum concentrations.**
(JPG)

**S4 Fig. ALT concentrations of all individuals and the mean concentration of each group with 95%CI on each day before maximum concentrations.**
(JPG)

**S5 Fig. Hematocrit of all individuals and the mean concentration of each group with 95% CI on each day after enrolment.**
(JPG)

**S6 Fig. Platelet count of all individuals and the mean concentration of each group with 95%CI on each day after enrolment.**
(JPG)

**S7 Fig. Hematocrit of all individuals and the mean concentration of each group with 95% CI on each day before the peak values.**
(JPG)

**S8 Fig. Platelet count of all individuals and the mean concentration of each group with 95%CI on each day before the peak values.**
(JPG)

**S1 Table. Results of the generalized estimating equation (GEE) examining the association of various laboratory results and body temperature with the treatment groups and days until peak.**
(DOCX)

## Acknowledgments

We thank Ouppatham Supasyndh, Research Unit, Phramongkutklao Hospital, and Data and Safety Monitoring Board members for study oversight and outcome review: Amnart Chaiprasert, Bancha Satirapoj, Sirakarn Tejavanija, Wichai Santimaleeworagun, Chantima Pongsuparbchon, Sujittra Suriwong, and Rattanawan Dispan. We thank Louise Kreitmann for study statistics consultation; all nurses and other health care personnel in all study sites and the internal medicine residents and fellows at the trial sites for their assistance with recruitment.

## Author Contributions

**Conceptualization:** Nattapat Nitinai, Vasin Vasikasin.

**Data curation:** Worapong Nasomsong, Worayon Chuerboonchai, Akarawin Tweekittikul, Thananut Bangchuad.

**Formal analysis:** Vasin Vasikasin.

**Funding acquisition:** Vasin Vasikasin.

**Investigation:** Nattapat Nitinai, Worapong Nasomsong, Worayon Chuerboonchai, Akarawin Tweekittikul, Vutthikorn Khingmontri, Bawornnan Panuvatvanich, Thananut Bangchuad, Maynanchaya Pongpraijaroen, Tanapol Roongfa-ngarm.

**Methodology:** Nattapat Nitinai, Worapong Nasomsong.

**Project administration:** Worapong Nasomsong, Worayon Chuerboonchai, Akarawin Tweekittikul.

**Resources:** Vasin Vasikasin.

**Supervision:** Vasin Vasikasin.

**Validation:** Vasin Vasikasin.

**Writing – original draft:** Nattapat Nitinai.

**Writing – review & editing:** Nattapat Nitinai, Worapong Nasomsong, Worayon Chuerboonchai, Akarawin Tweekittikul, Vutthikorn Khingmontri, Bawornnan Panuvatvanich, Thananut Bangchuad, Maynanchaya Pongpraijaroen, Tanapol Roongfa-ngarm, Vasin Vasikasin.

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
