## [Decision Letter · Decision Letter 0]

8 Jan 2024

Dear Dr Vasikasin,

Thank you very much for submitting your manuscript "Effect of montelukast in preventing dengue with warning signs among patients with dengue: a multicenter, randomized, double-blind, placebo-controlled trial" for consideration at PLOS Neglected Tropical Diseases. As with all papers reviewed by the journal, your manuscript was reviewed by members of the editorial board and by several independent reviewers. The reviewers appreciated the attention to an important topic. Based on the reviews, we are likely to accept this manuscript for publication, providing that you modify the manuscript according to the review recommendations. 

Sincerely,

Andrés F. Henao-Martínez, M.D.

Academic Editor

Michael Holbrook

Section Editor

Reviewer's Responses to Questions

**Key Review Criteria Required for Acceptance?**

**Methods**

-Are the objectives of the study clearly articulated with a clear testable hypothesis stated?

-Is the study design appropriate to address the stated objectives?

-Is the population clearly described and appropriate for the hypothesis being tested?

-Is the sample size sufficient to ensure adequate power to address the hypothesis being tested?

-Were correct statistical analysis used to support conclusions?

-Are there concerns about ethical or regulatory requirements being met?

Reviewer #1: The study design, being a multicenter, randomized, double-blind, placebo-controlled trial, is appropriate for investigating the efficacy of montelukast.

Clearly defined inclusion and exclusion criteria contribute to the study's rigor.

The duration of medication and follow-up is adequately justified.

Reviewer #2: The article describes in detail the clinical trial, which was conducted to determine the effect of montelukast in the prevention of dengue with alarming signs.

The clinical trial follows standard protocols for this type of study. For data analysis, standard tools were used, which is acceptable.

Reviewer #3: Yes. 

Incidence rate of transaminase levels in previous 2018 open-label retrospectively study could be cited as reference in this paper.

Data and safety monitoring board members composition that reviewed the study results could be added in the methods section.

**Results**

-Does the analysis presented match the analysis plan?

-Are the results clearly and completely presented?

-Are the figures (Tables, Images) of sufficient quality for clarity?

Reviewer #1: The presentation of participant enrollment and randomization is clear.

Descriptive statistics are well-organized and provide a comprehensive overview of the study population.

The primary and secondary outcomes are appropriately reported.

The use of Hazard Ratios and 95% confidence intervals for the primary outcome enhances result interpretation.

Subgroup analyses add depth to the findings, acknowledging potential variations.

Reviewer #2: Although the results are not desirable, a meticulous analysis is made of each of the possible factors that make the result.

Reviewer #3: Yes, safety results could be elaborated with graph/mean values showing change from admission of lab levels of hematocrit, platelets, transaminase levels after enrolment. As transaminase levels were defined as combination of AST and ALT concentrations of more than 3 times than normal, it would be better to include table or graph showing these levels change from baseline to clinical outcome.

Observed Transaminase levels were higher in Montleukast group, however no serious adverse event were reported. This should be presented in safety results with investigator causality assessment to study drug. 

Data safety and monitoring board recommendations following review of interim analysis could be added in the paper.

**Conclusions**

-Are the conclusions supported by the data presented?

-Are the limitations of analysis clearly described?

-Do the authors discuss how these data can be helpful to advance our understanding of the topic under study?

-Is public health relevance addressed?

Reviewer #1: The discussion effectively interprets the results in the context of prior studies, highlighting divergent findings.

The mention of the lower-than-expected incidence of warning signs is appropriately addressed, considering participant demographics.

The observed reduction in transaminase elevations is intriguing and warrants further exploration, as appropriately emphasized.

The limitations are acknowledged, maintaining transparency about potential biases.

The conclusion provides a concise summary and appropriately suggests avenues for future research.

inor

Reviewer #2: I agree that the difference in transaminase values does not necessarily have to do with the use of montelukast, and may be influenced by the patient's previous health status (sedentary lifestyle, nutrition, etc) to contracting the virus.

Reviewer #3: Yes.

**Editorial and Data Presentation Modifications?**

Reviewer #1: minor revision

Reviewer #2: It is recommended to increase the font size of the axes in figure 2.

Reviewer #3: Minor revision

**Summary and General Comments**

Reviewer #1: good

Reviewer #2: In a personal opinion, it is not recommended to use e-mail for the traffic of sensitive information. For future trials, the use of software with encrypted data is recommended.

The highest incidence strain in the trial was DENV-1 What would be the justification for it not being relevant at the time of the trial?

Reviewer #3: (No Response)

PLOS authors have the option to publish the peer review history of their article (what does this mean?). If published, this will include your full peer review and any attached files.

Reviewer #1: No

Reviewer #2: Yes: Javier Armando Gutierrez

Reviewer #3: No

Figure Files:

Data Requirements:

Reproducibility:

References

---

## [Editor Report · Decision Letter 1]

20 Jan 2024

Dear Dr Vasikasin,

We are pleased to inform you that your manuscript 'Effect of montelukast in preventing dengue with warning signs among patients with dengue: a multicenter, randomized, double-blind, placebo-controlled trial' has been provisionally accepted for publication in PLOS Neglected Tropical Diseases.

Best regards,

Andrés F. Henao-Martínez, M.D.

Academic Editor

Michael Holbrook

Section Editor

---

## [Editor Report · Acceptance letter]

25 Jan 2024

Dear Dr Vasikasin,

We are delighted to inform you that your manuscript, "Effect of montelukast in preventing dengue with warning signs among patients with dengue: a multicenter, randomized, double-blind, placebo-controlled trial," has been formally accepted for publication in PLOS Neglected Tropical Diseases.

Best regards,

Shaden Kamhawi

co-Editor-in-Chief

Paul Brindley

co-Editor-in-Chief
